# Assessment of Subclinical Renal Glomerular and Tubular Dysfunction in Children with Beta Thalassemia Major

**DOI:** 10.3390/children8020100

**Published:** 2021-02-03

**Authors:** Asmaa A. Mahmoud, Doaa M. Elian, Nahla MS. Abd El Hady, Heba M. Abdallah, Shimaa Abdelsattar, Fatma O. Khalil, Sameh A. Abd El Naby

**Affiliations:** 1Department of Pediatrics, Faculty of Medicine, Menoufia University, Shebin Elkom 32511, Egypt; doaaelian@yahoo.com (D.M.E.); drnono23@yahoo.com (N.M.A.E.H.); samehabdallah75@yahoo.com (S.A.A.E.N.); 2Department of Pediatrics, College of Medicine, King Faisal University, Al-Ahsa 31982, Saudi Arabia; 3Department of Clinical Pathology, National Liver Institute, Menoufia University, Shebin Elkom 32511, Egypt; dr.hebaabdallah2020@gmail.com; 4Department of Clinical Biochemistry and Molecular Diagnostics, National Liver Institute, Menoufia University, Shebin Elkom 32511, Egypt; shimaa.abdelsattar@liver.menofia.edu.eg; 5Department of Clinical and Molecular Microbiology and Immunology, National Liver Institute, Menoufia University, Shebin Elkom 32511, Egypt; fatma.khalil@liver.menofia.edu.eg

**Keywords:** cystatin C, N-acetyl beta-D-glucosaminidase (NAG), kidney injury molecule 1 (KIM-1), beta thalassemia major, iron overload

## Abstract

Background: A good survival rate among patients with beta thalassemia major (beta-TM) has led to the appearance of an unrecognized renal disease. Therefore, we aimed to assess the role of serum cystatin-C as a promising marker for the detection of renal glomerular dysfunction and N-acetyl beta-D-glucosaminidase (NAG) and kidney injury molecule 1 (KIM-1) as potential markers for the detection of renal tubular injury in beta-TM children. Methods: This case-control study was implemented on 100 beta-TM children receiving regular blood transfusions and undergoing iron chelation therapy and 100 healthy children as a control group. Detailed histories of complete physical and clinical examinations were recorded. All subjected children underwent blood and urinary investigations. Results: There was a significant increase in serum cystatin-C (*p* < 0.001) and a significant decrease in eGFR in patients with beta-TM compared with controls (*p* = 0.01). There was a significant increase in urinary NAG, KIM-1, UNAG/Cr, and UKIM-1/Cr (*p* < 0.001) among thalassemic children, with a significant positive correlation between serum cystatin-C, NAG and KIM-1 as regards serum ferritin, creatinine, and urea among thalassemic patients. A negative correlation between serum cystatin-C and urinary markers with eGFR was noted. Conclusion: Serum cystatin-C is a good marker for detection of glomerular dysfunction. NAG and KIM-1 may have a predictive role in the detection of kidney injury in beta-TM children.

## 1. Introduction

Beta thalassemia is considered the most common heterogeneous genetic disorder, resulting from reduced or absent beta globin synthesis, leading to globin chain imbalance [1]. Beta thalassemia is characterized by reduced hemoglobin production with excess α-goblins. This leads to microcytic hypochromic anemia, which is associated with oxidative stress and ineffective erythropoiesis, leading to chronic hemolytic anemia [2,3]. Beta thalassemia includes beta thalassemia major (beta-TM), beta thalassemia intermedia, or thalassemia minor [4]. Despite the increased survival of patients with thalassemia, some complications have been detected, such as renal injury or damage [5].

Hypoxia and chronic anemia lead to oxidative stress and lipid peroxidation, resulting in tubular cell function impairment [6]. Moreover, iron overload has an important role in the pathogenesis of kidney injury in thalassemic patients [7]. In addition, iron chelator toxicity can result in glomerular dysfunction. Hepatitis B or C infections may lead to a decrease in the glomerular filtration (GFR); moreover, hepatic and cardiac dysfunction caused by iron overload may result in renal impairment [8].

The deposition of hemosiderin in proximal and distal tubules can lead to interstitial fibrosis, tubular necrosis, and cortical atrophy. Injured tubules release cytotoxines and growth factors that result in tubulo-interstitial fibrosis and glomerular sclerosis [7].

Cystatin-C is filtered by the glomerulus freely due to its small size and is not secreted by the renal tubules but is reabsorbed and broken down by renal tubules. Therefore, it is considered an excellent marker of the glomerular filtration rate [9].

N-acetyl beta-D-glucosaminidase (NAG) is a lysosomal enzyme found in proximal renal tubules. An abnormal urinary NAG excretion was found in acute kidney injury, glomerulonephritis, nephrotic syndrome, and drug nephrotoxicity [10].

Kidney injury molecule-1 (KIM-1) is a trans-membrane glycoprotein; its level is related to tubular injury degree, inflammation in injured kidneys, urinary tract infection, and interstitial fibrosis [11]. Urinary KIM-1 is an essential biomarker for the early detection of renal tubular injury, as it is increased a few hours after acute kidney injury, before the increase in serum creatinine [12]. 

Therefore, we aimed to evaluate and assess the role of serum cystatin-C as a promising marker for the detection of renal glomerular dysfunction and of urinary kidney injury molecules (NAG and KIM-1) as potential markers for detection of renal tubular injury in beta thalassemia major children.

## 2. Subjects and Methods

### 2.1. Design

A case-control study was implemented on 100 children with beta thalassemia major 38 girls and 62 boys, with a mean age of (9.58 ± 4.07 years), and 100 healthy children matched in terms of age, gender, residence, and socioeconomic standard as a control group. This study was carried out from August 2019 to July 2020. They were recruited from an Egyptian pediatric hematology center (Menoufia Governorate, the place of the study). Prior to blood sample collection, written informed consent, approved from the Ethics Committee of Faculty of Medicine, Menoufia University (ID: 29/7/2019.PED), was given by all participants in the study and their guardians, who were informed about the main aim of the study, its benefits, and the absence of any risk. Patients were subjected to a detailed history, a complete physical and clinical examination, and laboratory investigations that included the following information: complete blood count, serum ferritin, creatinine, urea, sodium, potassium, and calcium by ELISA and urinary sodium, potassium, calcium, and uric acid by an enzymatic colorimetric method.

### 2.2. Diagnostic Inclusion and Exclusion Criteria

#### 2.2.1. Inclusion Criteria

Children with beta-TM were diagnosed by hemoglobin electrophoresis. Beta-TM patients of both sexes needed a blood transfusion every 2 to 3 weeks to maintain their hemoglobin levels at 8 gm/dL and were treated with iron chelation therapy using oral deferasirox. They did not have viral hepatitis B or C infection. Healthy controls had no hematological or renal diseases.

#### 2.2.2. Exclusion Criteria

Patients with diabetes, hypertension, failure to thrive, refractory rickets, hypertension, and primary renal disease were excluded from the study.

### 2.3. Sample Collection and Assay

Samples were collected by sterile venipuncture and divided into three parts. The first part was transferred into dipotassium ethylene diaminetetraacetic acid (EDTA) tube for complete blood count. The second part was transferred into a plain vacutainer tube, left to clot, and then centrifuged to separate serum used for the assessment of kidney function tests, electrolytes, and ferritin levels. Complete blood count was determined using an automated Sysmex XN-1000 hematology analyzer. Kidney function tests were performed using an AU680 chemistry analyzer. A serum ferritin assay was performed using a mini VIDAS immune-analyzer. Serum cystatin-C was measured by a quantikine human cystatin-C immunosorbent assay (ELISA) kit (Cat. No. DSCTC0; R&D systems, Inc., Minneapolis, MN, USA).

Fresh first morning midstream urine samples were collected in a sterile polypropylene container for analysis of creatinine, total protein, calcium, sodium, potassium, uric acid, and urinary markers including NAG and KIM-1.

For urinary KIM-1, centrifugation occurred for 20 min at 2000–3000 r.p.m. twice with the removal of supernatant. If precipitation appeared again, the sample was analyzed by an available quantitative sandwich immunoassay technique (Sun Red Biotechnology Company, Shanghai, China). Normalized KIM-1 level was obtained by dividing it by urine creatinine. 

Urinary N-acetyl beta-D-glucosaminidase was analyzed by colorimetric assay using 3-cresolsulfonphthaleyn-N-acetyl-beta-D-glucosaminide that was hydrolyzed by NAG with 3-cresolsulfonphthaleyn sodium salt release, which is determined photometrically at 580 U/g/Creatinine [13].

Urinary sodium, potassium, and calcium were analyzed by an enzymatic colorimetric method using an Integra 800 device. The urinary calcium to creatinine ratio and uric acid were measured by spectrophotometry.

Estimated glomerular filtration rate (eGFR) calculation was done by the modified Schwartz formula for children: eGFR (mL/min/1.73 m^2^) = height (cm) × 0.413/serum creatinine (mg/dL)(1)

A normal value of eGFR was ≥90 mL/min/1.73 m^2^ and the decreased value of eGFR was <90 mL/min/1.73 m^2^ [14].

The random urine UA/CR ratio was calculated as (random urine uric acid)/(random urine Cr) [15].

#### 2.3.1. Sample Size Calculation

The sample size relied upon 95% CI with 80% power, using an unpaired *t*-test and assuming an α (two-sided) of 0.05. Based on a previous study, the mean of the urinary NAG of β-thalassemia major case group was 106.5, while that of the healthy control group was 66.3, and the SD was 101.5, with a group size ratio of 1/1 [5].

#### 2.3.2. Statistical Analysis

Data were analyzed using IBM SPSS statistics version 20 (SPSS Inc., Chicago, IL). The chi-square test was used to examine the relation between qualitative variables. For quantitative data, comparison between two groups was done using either a Student’s *t*-test or a Mann–Whitney test (non-parametric *t*-test), as appropriate. Pearson’s correlation coefficient or the Spearman-rho method (as appropriate) was used to test the correlation between numerical variables. Multiple linear regression was applied to determine the relationship between dependent and independent variables. A *p*-value < 0.05 was considered significant.

## 3. Results

Demographic and clinical features are given in Table 1. No significant difference in age and sex were found between the two groups. There were significant differences in hemoglobin (Hb), hematocrit (HCT), and mean corpuscular volume (MCV) levels of two groups, as they were lower in patients with beta-TM than controls (*p* < 0.001). Serum ferritin was significantly higher in patients with beta-TM than in the controls (*p* < 0.001). Twenty-three percent of the studied patients underwent splenectomy (Table 1). Analysis of kidney function tests showed a significant increase in serum urea and creatinine in beta-TM patients compared with the controls (*p* < 0.001), but they were within the normal range for both groups, and a significant decrease in eGFR in patients with beta-TM compared with the controls (*p* = 0.01). Eleven percent of beta-TM had a low eGFR < 90. Moreover, there was no significant difference in serum sodium and calcium between the two groups, and there was a significant increase in serum potassium in patients with beta-TM than in the controls, but it was still within the normal range. In addition, analysis of urinary electrolytes and urinary protein whereby creatinine was divided, there was a significant increase in urinary uric acid/Cr, UCa/Cr, UProtein/Cr, and UK/Cr in patients with beta-TM than in the controls (Table 2). There was a significant increase in serum csytatin-C and urinary NAG, KIM-1, UNAG/Cr, and UKIM-1/Cr among thalassemic patients compared with the controls (*p* < 0.001) (Table 3). There was a significant positive correlation between serum cytatin C, UNAG, and UKIM-1 as regards blood transfusion index per year, serum ferritin, serum creatinine, and serum urea among patients with beta-TM (*p* < 0.001). In addition, there was a negative correlation between cystatin C, urinary markers and eGFR. Moreover, there was a significant positive correlation between UNAG, UKIM-1, UNAG/Cr, and UKIM-1/Cr as regards urinary protein/Cr, UCa/Cr, UNa/Cr, UK/Cr and Uuric acid/Cr (*p* <0.001). No significant correlation was found between renal markers as regards age, body weight and the time from diagnosis with beta-TM. There was a significant positive correlation between UKIM-1 and the iron chelation therapy with deferasirox (*p* < 0.001). Multiple linear regression revealed that serum cystatin-C, UNAG, and KIM-1 were independent risk factors for the occurrence of renal affection among beta-TM patients (Table 4 and Table 5). There was a significant increase in UNAG, UNAG/Cr, UKIM-1/Cr, serum creatinine, Uprotein/Cr, and UCa/Cr and a significant decrease in eGFR in beta-TM patients who underwent splenectomy compared to those who had splenomegaly, as shown in Table 6.

## 4. Discussion

In beta thalassemia syndromes, iron overload due to regular blood transfusion leads to iron deposition in renal proximal tubules, glomeruli, and interstitium, resulting in tubular atrophy, glomerulosclerosis, and interstitial fibrosis [16].

Analytical interpretation of the current study results showed a significant elevation of serum ferritin and serum creatinine in beta thalassemia major patients. Our patients are on regular deferasirox as an iron chelation therapy. Ponticelli et al. [17] demonstrated that renal involvement in children with beta-TM was caused by chronic anemia, increased iron deposition, and iron chelator toxicity. Vichinsky [18] found that deferasirox increases proteinuria and serum creatinine and may cause renal failure and found a reversible non-progressive elevation of serum creatinine in about 14% of beta-TM patients. 

The current study results showed decreased estimated GFR in eleven percent of thalassemic patients (*n* = 11). Glomerular capillary wall stretching, and subsequent endothelial and epithelial injury, can induce the transudation of macromolecules into the mesangium and glomerular dysfunction, which may cause a progressive decline in GFR [19]. Bekhit et al. [20] showed that the serum creatinine was higher in 40% of thalassemic compared with controls, but within a normal range, while the GFR was significantly lower in thalassemic patients compared with the controls. In addition, other studies found classical kidney function impairment (elevated serum creatinine and decreased eGFR) and early glomerular function impairment markers in children with beta-TM compared with controls [21,22]. On the other hand, some studies did not illustrate any difference in kidney function and GFR [23,24].

The present study showed hypercalciuria and an elevated urinary uric acid/creatinine ratio and urinary protein/creatinine ratio in children with beta-TM. Many studies have reported hypercalciuria, which was tested by the urine calcium to creatinine ratio, which is consistent with proximal tubulopathy and correlates with blood transfusion burden and deferasirox as iron chelation therapy [25,26,27]. Aldudak et al. [28] illustrated an increased level of urinary protein/creatinine ratio in children with beta-TM. Bekhit et al. [20] demonstrated increased urinary calcium in 26% of patients and increased urinary uric acid in 38% of patients. This may be explained by a decreased reabsorption of filtered uric acid from damaged renal tubules, combined with rapid erythrocyte turnover and excess uric acid urinary excretion [21,28]. 

Analytical interpretation of urinary kidney markers of the current results showed a significant increase in UNAG, UNAG/Cr, UKIM-1, and UKIM-1/Cr. Tantawy et al. [29] reported that 58.1% of patients with beta thalassemia had elevated levels of urinary NAG. Smolkin et al. [30] showed an increased urinary NAG and UNAG/Cr ratio in patients with β thalassemia major and intermedia. In addition, some studies have demonstrated an increase in urinary NAG [31,32]. Şen et al. [5] showed a significant increase in UNAG/CR ratio in patients with beta thalassemia compared with controls, but no significant differences were reported in urinary KIM-1-to-creatinine (UKIM-1/Cr). 

The current study showed a significant positive correlation between UNAG, UKIM-1, UNAG/Cr, and UKIM-1/Cr as regards serum ferritin, serum creatinine, and serum urea among patients with beta-TM. In addition, there was a negative correlation between urinary markers and eGFR. Moreover, there was a significant positive correlation between uNAG, uNAG/Cr, and uKIM-1/Cr as regards urinary protein/Cr, U Ca/Cr, UNa/Cr, and UK/Cr. Şen et al. [5] illustrated a significant positive correlation between urinary biomarkers as regards the UCa/Cr, UNa/Cr, UK/Cr and Uuric acid/Cr as urinary markers and urinary solutes represent the renal tubular functions. Nafea et al. [33] reported that the UKIM-1/Cr level was significantly higher in thalassemic patients on deferasirox therapy than patients on deferoxamine and deferiprone therapy. Dou et al. [34] demonstrated that thalassemic patients who took deferasirox were more likely to have increased serum creatinine. Al-khabori et al. [35] illustrated the nephrotoxicity of deferasirox in 10% of children with beta-TM patients who discontinued deferasirox, as it caused a persistent elevation in serum creatinine. Deferasirox nephrotoxicity was observed in 1 of 10 patients. Nephrotoxicity may be in the form of tubulopathy, glomerulonephritis, interstitial nephritis, and even renal failure [36]. Sánchez-González et al. [37] reported that deferasirox administration leads to partial necrosis in renal tubules and increased UKIM-1, protein, and glucose secretion. Martin-Sanchez et al. [38] illustrated that deferasirox had a direct toxic effect on tubular cells and induced mitochondrial dysfunction. Balocco et al. [39] illustrated that the alternate use of deferasirox and deferiprone allowed for drug tolerability with no adverse effects and a similar efficacy in reducing serum ferritin. Ponticelli et al. [17] showed that the deferasirox dose should be decreased by 10 mg/kg if serum creatinine rises 33% above pretreatment values and above the age-appropriate upper limit of normal at two consecutive visits. On the other hand, Aldudak et al. [28] reported normal serum creatinine in patients receiving deferasirox therapy. Our study showed a significant positive correlation between serum cyatatin C, UNAG, UKIM-1 and the blood transfusion index per year among patients with beta-TM. Behairy et al. [40] found a significant positive correlation between serum cyatatin C and the frequency of blood transfusion per year in patients with beta-TM. We found no significant correlation between renal markers and body weight among patients with beta-TM. Baxmann et al. [41] showed no significant correlation between serum cystatin C as regards body weight, body muscle cell and fat-free mass. 

The present study showed a significant increase in serum creatinine, Uprotein/Cr, UCa/Cr, UNAG, UNAG/Cr, and UKIM-1/Cr and a decrease in eGFR in β-TM patients who underwent splenctomy. Tantawy et al. [29] illustrated a prominent elevation of NAG in splenectomized patients and proved that splenectomy was an independent risk factor for renal tubular abnormalities. Ongazyooth et al. [42] found that tubular defects were more common in splenectomized patients. In addition, Bekhit et al. [20] reported higher levels of urinary NAG in splenectomized patients than in patients with splenomegaly. Belhoul et al. [43] stated that, after splenectomy, the transaminases were higher and serum albumin was lower compared with nonsplenectomized patients. This may be explained by the fact that the rates of iron-overload-related organ damage in splenectomized patients are higher than those in non-splenectomized patients, as the primary underlying pathology of red cell dysfunction persists after splenectomy. Ismail et al. [44] showed an elevated serum creatinine and Tantawy et al. [29] found an elevation of total urinary protein in splenectomized patients than in patients with splenomegaly. 

Our results showed a significant increase in serum cystatin-C among beta-TM patients, with a significant positive correlation with serum ferritin and creatinine and a significant negative correlation with eGFR. Economou et al. [26] found that 36% of beta-TM patients had increased serum cystatin-C. Moreover, there was elevated serum cystatin-C in beta-TM patients with or without chelation therapy, and there was a significant positive correlation with serum creatinine and a negative correlation with eGFR [21]. Ali and Mahmoud [45] showed significantly higher levels of serum cystatin-C, with a significant positive correlation as regards serum creatinine and serum ferritin in children with beta-TM compared with a control group and a significant, strong negative correlation between serum cytatin-C and eGFR. Furthermore, Elbedewy et al. [46] showed a significant negative correlation between serum cystatin C and eGFR in adult patients with beta-TM.

Afshan et al. [47] found a significant moderate negative correlation between kidney T2* relaxation time by magnetic resonance imaging of kidneys (MRI) and the serum ferritin in patients with beta-TM. Therefore, MRI T2* may estimate the renal iron burden in thalassemic patients.

Strengths of the current study: we have analyzed the relationship between urinary kidney injury molecules and regular renal parameters such as serum ferritin, serum creatinine, and serum urea. Furthermore, we showed that an increase in UNAG, UNAG/Cr, UKIM-1/Cr, and Uprotein/Cr was found in beta-TM with splenectomy. Glomerular dysfunction was evaluated by serum cystatin-C, the glomerular filtration rate, serum creatinine, and the urinary protein creatinine ratio.

Recommendation: A serial assay of UNAG should be measured to evaluate its possible prognostic value in AKI in beta-TM patients. 

## 5. Conclusions

Patients with beta thalassemia are at high risk of renal glomerular and tubular impairment, which are not found in routine renal investigations. Serum cystatin-C is a good predictive marker in the evaluation of glomerular dysfunction. Urine concentrations of NAG and KIM-1 represent sensitive, specific, and highly predictive early biomarkers for acute renal injury in patients with beta TM when subclinical kidney injury or dysfunction is expected before serum creatinine increases.

## Figures and Tables

**Table 1 children-08-00100-t001:** Demographic and clinical characteristics among studied groups.

Characteristics of Groups	TM Patients	Controls	*p*-Value
Age (year) Median (Range)	10 (2–18)	8 (2–17)	0.08
Sex			
Male	62 (62.0%)	56 (56.0%)	0.39
Female	38 (38.0%)	44 (44.0%)	
Hb (g/dL) Mean ± SD	7.17 ± 0.77	12.11 ± 0.83	<0.001 *
HCT (%) Mean ± SD	21.62 ± 2.42	36.32 ± 2.49	<0.001 *
MCV (fl) Mean ± SD	64.95 ± 13.85	72.55 ± 15.94	<0.001 *
Body weight (kg)	33.48 ± 13.28	36.3 ± 14.66	0.05 *
Ferritin (ng/mL) Mean ± SD	2820.55 ± 742.81	44.50 ± 13.14	<0.001 *

TM: Thalassemia Major; Hb: Hemoglobin; HCT: Hematocrit; MCV: Mean Corpuscular Volume. * Significant difference.

**Table 2 children-08-00100-t002:** Kidney function tests among studied groups.

Parameters	TM Patients	Controls	*p*-Value
Serum Creatinine (mg/dL) Mean ± SD	0.80 ± 0.13	0.55 ± 0.10	<0.001 *
Serum Urea (mg/dL) Mean ± SD	22.64 ± 3.42	13.40 ± 3.12	<0.001 *
eGFR (mL/min/1.73 m^2^) Mean ± SD	112.89 ± 17.33	118.50 ± 14.04	0.01 *
Urinary protein/Cr Median (Range)	0.29 (0.08–0.45)	0.12 (0.02–0.20)	<0.001 *
UCa/Cr Median (Range)	0.28 (0.08–0.45)	0.16 (0.05–0.22)	<0.001 *
UNa/Cr Median (Range)	2.55 (0.40–10.0)	2.75 (0.20–5.0)	0.18
UK/Cr Median (Range)	2.2 (0.90–4.0)	1.45 (0.50–3.0)	<0.001 *
U uric acid/Cr Median (range)	0.50 (0.2–1.43)	0.33 (0.14–1.12)	0.04 *

eGFR: Estimated glomerular filtration rate; Cr: Creatinine; UCa: Urinary calcium; UNa: Urinary sodium; UK: Urinary potassium. * Significant difference.

**Table 3 children-08-00100-t003:** Serum cystatin-C, urinary NAG, urinary KIM-1 among studied groups.

Parameters	TM Patients	Controls	*p*-Value
Urine NAG (U/g/Creatinine) Mean ± SD	29.84 ± 8.76	7.60 ± 1.81	<0.001 *
Urine KIM-1 (ng/g/Creatinine) Mean ± SD	5.05 ± 0.65	3.50 ± 0.74	<0.001 *
Serum Cystatin C (mg/L) Mean ± SD	1.23 ± 0.32	0.81 ± 0.06	<0.001 *
UNAG/Cr Median (range)	2.0 (0.10–2.90)	0.35 (0.05–0.70)	<0.001 *
UKIM-1/Cr Median (range)	0.16 (0.02–0.26)	0.03 (0.01–0.07)	<0.001 *

NAG: N-acetyl beta-D-glucosaminidase; KIM-1: Kidney injury molecule 1; UNAG: Urinary N-acetyl beta-D-glucosaminidase; UKIM-1: Urinary kidney injury molecule 1. * Significant difference.

**Table 4 children-08-00100-t004:** Univariate correlation and multi-linear regression between serum Cytatin-C, UNAG, or UKIM-1 and other parameters among studied beta-TM patients.

Parameters	Cystatin-C (mg/L)		UNAG (U/g/Creatinine)		UKIM-1 (ng/g/Creatinine)	
	r	Beta	r	Beta	r	Beta
Age (year)	−0.062	0.013	−0.038	0.089	−0.013	0.026
Body weight (kg)	−0.046	0.014	−0.050	0.017	−0.081	0.012
Ferritin (ng/mL)	0.519 *	0.001 *	0.781 *	0.009 *	0.307 *	0.001 *
Serum Creatinine (mg/dL)	0.659 *	0.003 *	0.474 *	−7.59 *	0.332 *	4.32 *
Serum Urea (mg/dL)	0.412 *	0.013	0.457 *	0.400	0.333 *	−0.034
eGFR (mL/min/73 m^2^)	−0.517 *	0.001	−0.351 *	0.074	−0.228 *	−0.002
Urinary protein/Cr	0.151	0.012	0.613 *	5.42	0.457 *	−0.456
UCa/Cr	0.099	0.015	0.524 *	5.26	0.332 *	−0.493
UNa/Cr	0.117	0.021	0.562 *	−0.075	0.409 *	0.013
UK/Cr	0.105	0.025	0.451 *	−1.43	0.221 *	0.183
Uuric acid/Cr	0.316	0.039	0.422 *	0.003	0.412 *	0.001
Iron chelation	−0.523	0.001	−0.062	−0.005	0.583 *	0.003
Transfusion index (mL/kg/year)	0.307 *	0.001	0.258 *	−0.009 *	0.237 *	0.003
Time from diagnosis (year)	−0.144	0.013	−0.114	−0.741	−0.005	0.021

* Significant difference.

**Table 5 children-08-00100-t005:** Univariate correlation and multi-linear regression between UNAG/Cr or UKIM-1/Cr and other parameters among studied beta-TM patients.

Parameters	UNAG/Cr		UKIM-1/Cr	
	r	Beta	r	Beta
Age (year)	0.037	−0.021	−0.042	0.001
Ferritin (ng/mL)	0.723 *	0.001 *	0.850 *	−0.005 *
Serum Creatinine (mg/dL)	0.286 *	−1.35 *	0.498 *	0.026 *
Serum Urea (mg/dL)	0.260 *	0.025	0.502 *	0.003
eGFR (mL/min/73 m^2^)	−0.255 *	0.003	−0.376 *	0.001
Urinary protein/Cr	0.622 *	0.241	0.685 *	0.080
UCa/Cr	0.503 *	0.646	0.580 *	0.058
UNa/Cr	0.601 *	−0.008	0.580 *	−0.005
UK/Cr	0.476 *	0.001	0.571 *	0.009
Uuric acid/Cr	0.432 *	0.008	0.409 *	0.003
Iron chelation	0.043	0.001	−0.044	−0.005
Transfusion index (mL/kg/year)	0.289 *	0.003	0.289 *	−0.005
Time from diagnosis (year)	0.041	0.001	−0.056	0.001

* Significant difference.

**Table 6 children-08-00100-t006:** Urinary NAG and KIM-1 among studied thalassemia major patients regarding splenectomy.

Parameters	Studied Thalassemia Major Patients		*p*-Value
	With Splenectomy (No. = 23)	Without Splenectomy (No. = 77)	
Urine NAG (U/g/Creatinine) Mean ± SD	42.04 ± 3.87	26.19 ± 6.09	<0.001 *
Urine KIM-1 (ng/g/Creatinine) Mean ± SD	5.24 ± 0.78	5.0 ± 0.60	0.19
UNAG/Cr Mean ± SD	2.37 ± 0.47	1.83 ± 0.49	<0.001 *
UKIM1/Cr Mean ± SD	0.20 ± 0.04	0.14 ± 0.04	<0.001 *
eGFR (mL/min/1.73 m^2^) Mean ± SD	103.7 ± 16.8	115.6 ± 16.6	0.003 *
Urinary protein/Cr Median (Range)	0.34 (0.18–0.45)	0.20 (0.08–0.42)	<0.001 *
UCa/Cr Median (Range)	0.32 (0.16–0.45)	0.19 (0.08–0.38)	<0.001 *
Serum Creatinine (mg/dL) Mean ± SD	0.89 ± 0.11	0.78 ± 0.12	<0.001 *

No: number. * Significant difference.

## Data Availability

Data available on request due to restrictions e.g. privacy or ethical.

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
