# Peer review of "Assessment of Subclinical Renal Glomerular and Tubular Dysfunction in Children with Beta Thalassemia Major"

_children, 2021, doi:10.3390/children8020100_

Round 1

Reviewer 1 Report

In this MS the authors report the results of a clinical investigation on tubular abnormalities in thalassemic children. The issue is interesting. I have however several concerns which are released to the authors as follows:

Major

-Did the authors studied incident or prevalent children? A CONSORT diagram may help the understanding of patients enrollment.

-the correlation between urinary molecules (UNAG, UKIM-1, UNAG/Cr 153 and UKIM-1/Cr) and serum creatinine, may be questionable, since children with different muscle mass were studied,

-The authors show also several other correlations, e.g. between UNAG, 156 UNAG/Cr and UKIM-1 an urinary uric acid, urinary protein/Cr, UCa/Cr, UNa, and 157 UK/Cr and between serum cystatin-C and serum 158 ferritin and creatinine that have meaning which are difficult to explain.

---In addition, there was significant negative correlation between serum cystatin-C and eGFR. This represents a tautology, since cystatinC is a marker of eGFR.

-serum creatinine, Uprotein/Cr and splenectomy data need to better discussed.

-Table 1. In place of the 2 columns indicating eGFR, I would suggest one single column reporting either mean+SD  or median (IOR) in the case of a non-Gaussian distribution.

-control subjects were borderline younger, suggesting that groups were not perfectly age-matched.

Table 1. Patients had higher serum creatinine, cystatine and urea, but similar eGFR vs. controls. Please check the data.

The authors need to include in their analyses the time from diagnosis.

Table 4 reports univariate analysis results. Multiple regression analysis needs also to be used.

In the study, it is unclear the role of chelating therapy on renal damgae.

-English is very poor.

Author Response

Comment: Did the authors studied incident or prevalent children? A CONSORT diagram may help the understanding of patient's enrollment.

Answer: we studied prevalent children and patient's enrollment through A CONSORT diagram and it was uploaded as a pdf figure.

Comment: The correlation between urinary molecules (UNAG, UKIM-1, UNAG/Cr 153 and UKIM-1/Cr) and serum creatinine may be questionable, since children with different muscle mass were studied.

Answer: We didn't measure the muscle mass, we only had body weight of patients. Therefore, there was no significant correlation between serum cystatin C and urinary markers as regard body weight and that was added to tale (4) and discussed at line 249-251. Muscle mass detection may be added to the recommendations.

Comment: -The authors show also several other correlations, e.g. between UNAG, 156 UNAG/Cr and UKIM-1 an urinary uric acid, urinary protein/Cr, UCa/Cr, UNa, and 157 UK/Cr and between serum cystatin-C and serum 158 ferritin and creatinine that have meaning which are difficult to explain.

Answer:  A significant correlations between urinary markers and urinary solutes were explained at lines 226-229. A significant correlations between serum cystatin C , as regard serum creatinine and ferritin were discussed at lines 273 and 274.

Comment: In addition, there was significant negative correlation between serum cystatin-C and eGFR. This represents a tautology, since cystatinC is a marker of eGFR.

Answer: Cystatin C is a marker of eGFR, we added all studies that showed a significant negative correlations between them at lines 270-277.

Comment: Serum creatinine, Uprotein/Cr and splenectomy data need to better discuss.

Answer: It was discussed at lines 264-266.

Comment: Table 2. In place of the 2 columns indicating eGFR, I would suggest one single column reporting either mean+SD or median (IOR) in the case of a non-Gaussian distribution.

Answer: It was modified to single column at table 2.

Comment: -Control subjects were borderline younger, suggesting that groups were not perfectly age-matched.

Answer: Median age of patients 50% was 10 years; median age of controls was 8 years but within the same range 2-18 years and 2-17 years with no significant difference.

Comment: Table 1. Patients had higher serum creatinine, cystatine and urea, but similar eGFR vs. controls. Please check the data.

Answer: Elven patients had low eGFR < 90 and that was mentioned at line 149.

Comment: The authors need to include in their analyses the time from diagnosis.

Answer: This was added at table 4.

Comment: Table 4 reports univariate analysis results. Multiple regression analysis needs also to be used.

Answer: Multiple linear regression analysis was used at tables 4 and 5 and added to the result section lines 166-168.

Comment: In the study, it is unclear the role of chelating therapy on renal damage.

Answer: That was added at table 4 and 5 and result analysis at line 164.

Comment: -English is very poor.

Answer: English editing was done by MDPI English editing with a certificate.

Reviewer 2 Report

This report suggests that cystatin-C, NAG, and KIM-1 can be used as markers when evaluating kidney function in patients with major thalassemia.
It is lacking in many ways and there are many parts to be supplemented.

Major
1. It has been mentioned several times that blood transfusions and iron chelating agents induce nephrotoxicity in patients with thalassemia major. In the Result section, there is no analysis of the relationship between the number of red blood cell transfusions, transfusion period or prevalence period or age, iron chelating agent use, drug dose, and renal glomerular markers.

2. The evidence for conclusion that NAG and KIM-1 play a predictive role in kidney injury in beta-TM children are weak.

Minor
1. Line 124. Base on previous study…
- what is previous study. Is it pilot study? If the result is published, record the reference. If the result is not published, the contents and results of the pilot study should be recorded in the method and result of this paper.

2. Table
A. In all tables, mean ± SD and median (range) are written together. Choose only one of the two values ​​and write it down.
B. I don't really understand what it means to record a test of significance. Why did the authors write the test of significance? If there is no specific reason, it is better to delete it.C. It would be better to modify Tables 1, 2, 3, 5 in the form below.

TM

Controls

p-value

Age (year)

9.58 ± 4.07

8.59 ± 4.56

0.08

Male, n (%)

62 (62.0)

56 (56.0)

0.39

Hb (g/dl)

7.17 ± 0.77

12.11 ± 0.83

<0.001

3. It is better to delete the "*significant difference" in the comment of the table.

There is already a statement that less than 0.05 is considered statistically significant in the method section.

4. The part written in bold in the table should be changed regularly.

5. It is better to delete p-value from Table 4 and display only r value. It's better to add *marks to statistically significant values.

Author Response

Comment 1. It has been mentioned several times that blood transfusions and iron chelating agents induce nephrotoxicity in patients with thalassemia major. In the Result section, there is no analysis of the relationship between the number of red blood cell transfusions, transfusion period or prevalence period or age, iron chelating agent use, drug dose, and renal glomerular markers.

Answer 1: We added the analysis of correlations between blood transfusion index, age from the diagnosis and iron chelation therapy and urinary markers at tables 4, 5. Blood transfusion index relationship was discussed at lines 245-248.

Comment 2. The evidence for conclusion that NAG and KIM-1 play a predictive role in kidney injury in beta-TM children is weak.

Answer 2: Multiple linear regression showed that serum cystatin C, UNAG and UKIM-1 were independent risk factors for renal affection in children with beta thalassemia at tables 4, 5.

Minor
Comment 1. Line 124. Base on previous study…
- What is previous study? Is it pilot study? If the result is published, record the reference. If the result is not published, the contents and results of the pilot study should be recorded in the method and result of this paper.

Answer 1: The previous study was published and the reference was added [5] at line 128.

Comment 2. Table
A. In all tables, mean ± SD and median (range) are written together. Choose only one of the two values ​​and write it down.
B. I don't really understand what it means to record a test of significance. Why did the authors write the test of significance? If there is no specific reason, it is better to delete it. C. It would be better to modify Tables 1, 2, 3, 5 in the form below.

TM

Controls

p-value

Age (year)

9.58 ± 4.07

8.59 ± 4.56

0.08

Male, n (%)

62 (62.0)

56 (56.0)

0.39

Hb (g/dl)

7.17 ± 0.77

12.11 ± 0.83

<0.001

 Answer 2: A: we used mean ± SD and median (range) according to the test of significance not mentioned together.

B: We deleted the test of significance from tables 1,2, 3, 6.

C: All tables 1, 2, 3, 6 were modified according to your comments.

Comment 3. It is better to delete the "*significant difference" in the comment of the table.

There is already a statement that less than 0.05 is considered statistically significant in the method section.

Answer 3: It was deleted.

Comment 4. The part written in bold in the table should be changed regularly.

Answer 4: It was changed.

Comment 5. It is better to delete p-value from Table 4 and display only r value. It's better to add *marks to statistically significant values.

Answer 5: p-value was deleted from table 4 and 5 and r value was displayed with adding *marks to statistically significant values.

Round 2

Reviewer 1 Report

The MS improved, although a huge amount of correlations is offered without much discussion.

In the conclusion the authors state:

Kidney magnetic resonance imaging 285 T2* (MRI T2*) can be used as a measure for the detection of hemosiderosis 

This statement seems to derive from the studies performed, although I was not able to find RM results. If so, this issue could be reported in the discussion, but not among the new data offered by the study.

Reviewer 2 Report

There are some insufficiency, but overall it is well revised. The whole manuscript need to revise the English
